# Potential of citizen science to advance urban planetary health research in low and middle-income countries: A scoping review

Amollo Ambole[1], Christer Anditi[1], Tolu Oni[2,3]*

1 Design Department, University of Nairobi, Nairobi, Kenya, 2 University of Cambridge, MRC Epidemiology Unit, Cambridge, United Kingdom, 3 UrbanBetter Academy Initiative, Future Africa, University of Pretoria, Pretoria, South Africa

* tolullah.oni@mrc-epid.cam.ac.uk

## Abstract

Planetary health has emerged as a transdisciplinary field to capture the interdependencies between environmental changes and human health. Nowhere is this more critical than in the low- and middle-income country (LMIC) settings where the majority of the world's population live. These settings are undergoing rapid urbanisation that could further threaten planetary boundaries. The collaborative and societally engaged nature of planetary health means more participatory and dynamic methods are needed to better characterise these exposures. Citizen science has the potential to enable the co-production of community-relevant evidence but the extent to which this is being deployed for planetary health in LMIC cities has not been synthesised. To synthesise evidence on the use of citizen science for planetary health-relevant studies in urban LMIC settings, we conducted a scoping review, following the Preferred Reporting Items for Systematic reviews and Meta-Analyses extension for Scoping Reviews and Joanna Briggs Institute's stages of conducting a scoping review. Inclusion criteria included empirical studies in LMICs, with a focus on cities and published in English within the last 10 years. Of the 31 eligible studies included, the majority focused on biodiversity, illustrating the unharnessed potential of deploying citizen science to advance understanding of a broader range of planetary health variables in LMIC cities. Our finding of a predominance of Global North funding for these studies highlights the need for greater diversity of funding sources and for a shift in the centre of gravity of funding decisions to optimise alignment of research priorities with contextual realities in the Global South. To inform future research, we propose a standardised reporting format for citizen science planetary health projects and guidelines to optimise data reliability and validity.

## Introduction

The Anthropocene, a proposed geological epoch, is characterised by the profound and pervasive impact of human activities on Earth's environmental systems in sharp

**Data availability statement:** All relevant data are within the paper.

**Funding:** This work was funded by a grant from the Fondation Botnar (Grant number IMG-22-010). TO is supported by the National Institute for Health Research (NIHR) (16/137/34) using UK aid from the UK Government to support global health research. The views expressed in this publication are those of the author(s) and not necessarily those of the NIHR or the UK Department of Health and Social Care. The funders had no role in study design, data collection and analysis, decision to publish, or preparation of the manuscript.

**Competing interests:** The authors have declared that no competing interests exist.

contrast to the preceding Holocene epoch that provided a stable and accommodating environment for human progress [1,2]. The Anthropocene can be closely linked to the concept of planetary boundaries, which is a framework that identifies key Earth system processes that must be maintained within safe limits to ensure the stability and resilience of the biosphere. Exceeding these nine planetary boundaries poses significant threats to both environmental and human health due to the risk of triggering abrupt and non-linear planetary-scale changes [3]. Recent assessments indicate that six of the planetary boundaries have already been transgressed, leading to potentially catastrophic consequences for ecosystems and human societies (Fig 1). To mitigate these risks, it is imperative to undertake urgent and coordinated global action to restore and maintain the boundaries within safe limits [4].

## Addressing planetary health through citizen science

To tackle the compounding impacts of the Anthropocene, planetary health is proposed as a transdisciplinary field in which human health and natural systems can be simultaneously safeguarded [5]. As defined by the Lancet Commission on planetary health, "planetary health is the achievement of the highest attainable standard of health, wellbeing, and equity worldwide through judicious attention to the human systems—political, economic, and social—that shape the future of humanity and the Earth's natural systems that define the safe environmental limits within which humanity can flourish." [6]. By focusing on research collaboration, data integration, and community engagement, the planetary health framework can address the interconnected challenges of environmental degradation and human well-being resulting from the Anthropocene. Fig 2 illustrates the compounding effects of these interconnected challenges by showing how the escalation of human pressures such as climate change, ozone depletion, biodiversity loss, and urbanization, on the global environment leads to environmental changes and ecosystem impairments, which in turn lead to poor outcomes for human health and well-being (ibid).

Advancing planetary health necessitates the rigorous monitoring of interactions between environmental changes and human health outcomes. Citizen science, which is a participatory approach of including non-scientists in research, has emerged as a vital tool that enables the collection of large datasets and production of community-relevant evidence [7–9]. Initially recognized for its role in biodiversity and habitat monitoring, citizen science has increasingly been applied to diverse environmental and human health research projects [10]. Various definitions, mostly from the Global North, demonstrate the evolving nature of citizen science, with terms such as non-professional scientists, volunteers, non-institutional partners, and non-scientific stakeholders emerging as synonyms for citizen scientists (ibid).

The role of citizen science in improving urban environments is an emerging subfield that emphasises how volunteers can enhance environmental and human health in their cities [11–13]. However, in the context of the Global South, citizen science remains less visible and underutilised due to resource constraints, infrastructure limitations, cultural and political barriers, and skill gaps [14,15]. Despite these challenges, we argue that citizen science in the Global South is essential for addressing

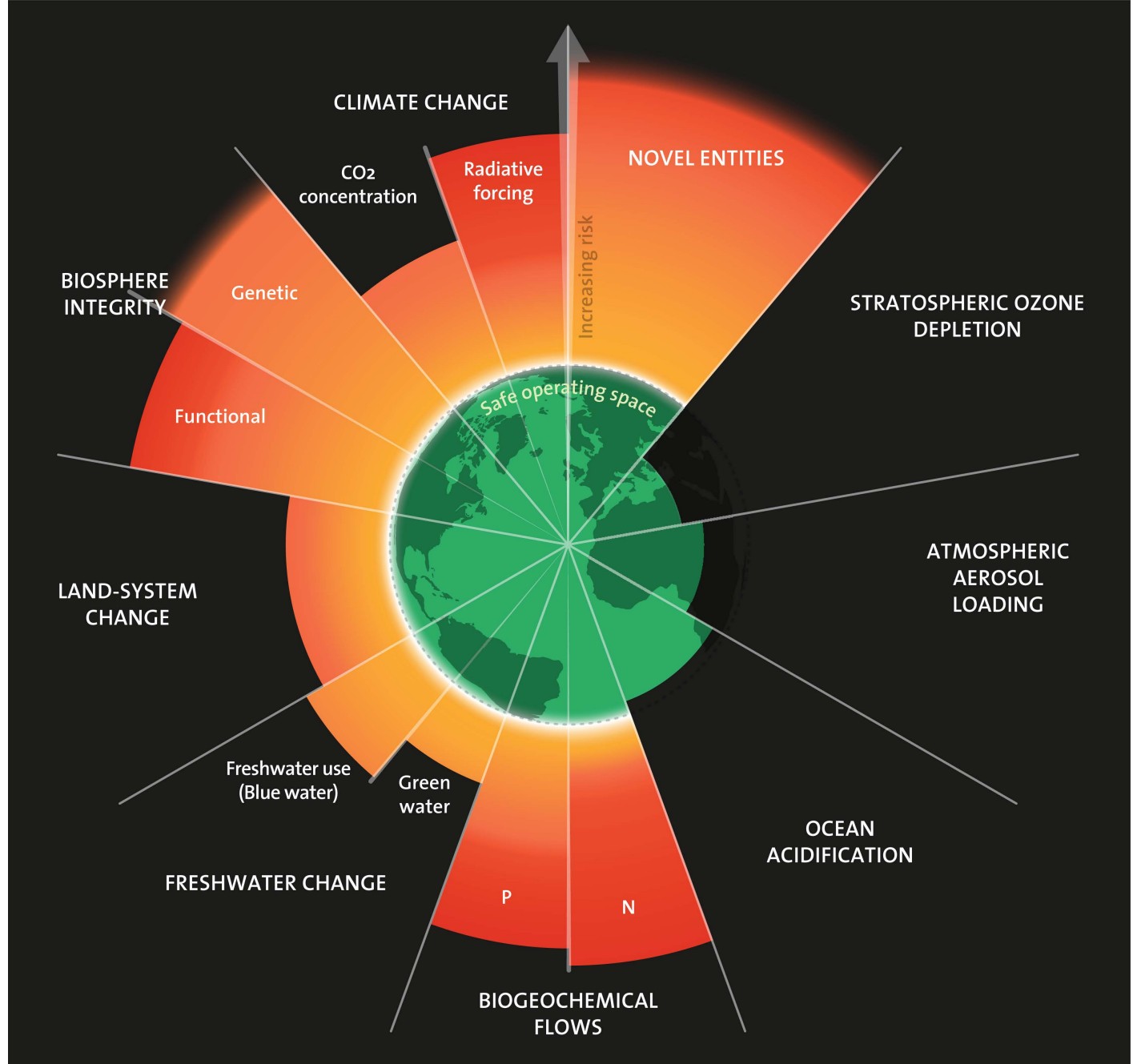

**Fig 1. Current status of the nine planetary boundaries (Source: Azote for Stockholm Resilience Centre).**

the severe impacts of environmental degradation in these regions [16], especially in cities in low- and middle- income countries (LMICs), where the most rapid urbanisation is occurring.

Several initiatives have contributed to an improved understanding of how citizen science is being advanced in LMICs. For example, the *UrbanBetter Cityzens* initiative engages young people in African cities in monitoring air quality and other urban health indicators [17]. Similarly, Stanford's *Our Voice* project promotes citizen science research for

**Fig 2. Harmful effects of ecosystem change on human health (Source: [6]).**

health equity around the world including in LMICs [18]. In Argentina, a government-led mapping exercise of 30 citizen science projects reveals a rich ecosystem for citizen science collaborations that co-create innovative solutions, inform public policies, and promote behavioural change [19]. The *Regreening Africa* report also highlights the benefits of citizen science in the use of the Regreening app that collected data from 180,000 farmers across Africa to promote tracking of land restoration efforts [20]. Furthermore, regional associations and networks such as the *African Citizen Science Association*, *Brazilian Citizen Science Network* and *Citizen Science India* demonstrate the growing interest in citizen science in LMICs.

Given this momentum, a better understanding of how citizen science can generate more community relevant data and solutions is crucial. This study thus explores the intersections between citizen science and planetary health to unpack the potential for citizen science to enhance planetary health in LMICs. Our specific objectives were to conduct a scoping review of empirical studies on citizen science to identify the methodologies and outcomes relevant to planetary health in urban settings of LMICs. We further sought to explore lessons and challenges identified in the reviewed studies to inform recommendations for the future.

## Methods

This study sits at the nexus of two main bodies of knowledge, - citizen science and planetary health, - both of which have diverse interpretations and applications [6,10]. To synthesise the diversity of work in this nexus field coherently requires an exploratory approach that can generate hypotheses on how citizen science and planetary health interlink in the urban settings of LMICs. As advised by Pollock and co-workers, hypothesis-generating is best done through a scoping review as

opposed to a systematic review that is better suited for hypothesis-testing [21]. Moreover, a scoping review is appropriate for presenting broad overviews in emerging or complex fields that have yet to be reviewed comprehensively [22]. In this study, the nexus between citizen science and planetary health has yet to be reviewed comprehensively within the context of urban settings of LMICs, necessitating this scoping review.

This scoping review adopted a Preferred Reporting Items for Systematic reviews and Meta-Analyses extension for Scoping Reviews (PRISMA-ScR) (S1 Table) and Joanna Briggs Institute's (JBI) six stages of conducting a scoping review [22–24]. The JBI six stages have been refined to include five stages of i) identifying the research questions, ii) identifying relevant studies by balancing contextual relevance with applicability, iii) selecting studies using an iterative team approach to arrive at a coherent understanding of the nexus field, iv) charting the data using a data extraction template, v) collating, summarising, and reporting results including implications for practice and advocacy. The sixth recommended stage of consulting with stakeholders will be done in a separate study given the diversity of experts working across the nexus of citizen science and planetary health in LMICs.

In line with stage one above, the research questions guiding this scoping review are:

• What are the methodologies used, and what human/planetary health outcomes are measured, in citizen science projects that focus on promoting planetary health in urban LMIC settings?

• What are the lessons and challenges in the studies that could inform enhanced practices for initiating and sustaining citizen science projects to promote planetary health in LMICs?

For stage two and three, we identified keywords and their synonyms from an initial scoping of literature at the nexus of citizen science and planetary health. We developed the search strings from the keywords of citizen science, planetary boundaries and climate change, urbanisation, and LMICs. The search strings were used in SCOPUS and Web of Science (WoS) as shown in S2 Table. Citizen science was used as a keyword to increase the likelihood of finding empirical studies. Two independent reviewers screened titles and abstracts separately using the inclusion criteria of empirical studies, published in English within the last 10 years. Reviews and editorial were excluded. After the screening of abstracts, the three reviewers agreed on the eligibility criteria and the expansion of the geographical distribution of the articles across the Global South using Google Scholar. Throughout the process, the three reviewers used an iterative team approach to arrive at a shared coherent understanding of the nexus field that resulted in the 31 academic articles of this scoping review (Fig 3). The grey literature that emerged in the fringes of this search were not included in the scoping review as they did not present substantial empirical material for study. They were nevertheless discussed and cited in the introductory chapter of this paper under the subsection on addressing planetary health through citizen science.

In stage four, a data extraction template (S3 Table) was developed to chart key information from each empirical study following the JBI methodology for scoping reviews [22]. Table 1 provides a theoretical framework for the review that relates the research questions to the extracted data and discussion (Table 1). The results were synthesised in stage five through thematic analyses and narrative summaries, focusing on methodologies, outcomes, challenges, gaps, and recommendations as discussed in the next section.

## Results

### 1. Background and study characteristics

A total of 31 studies were included in the review. While all studies included urban contexts (as per inclusion criteria), some studies focused on both rural and urban areas (Table 2). For foundational context, we extracted data related to the lead institutions, their funding sources, the geographical locations of the studies, and the topics investigated. Given the Global South focus of this paper, we also extracted data on the distribution of funding sources and institutional dominance between the Global North and the Global South. We used the institution of the corresponding author as a proxy for the

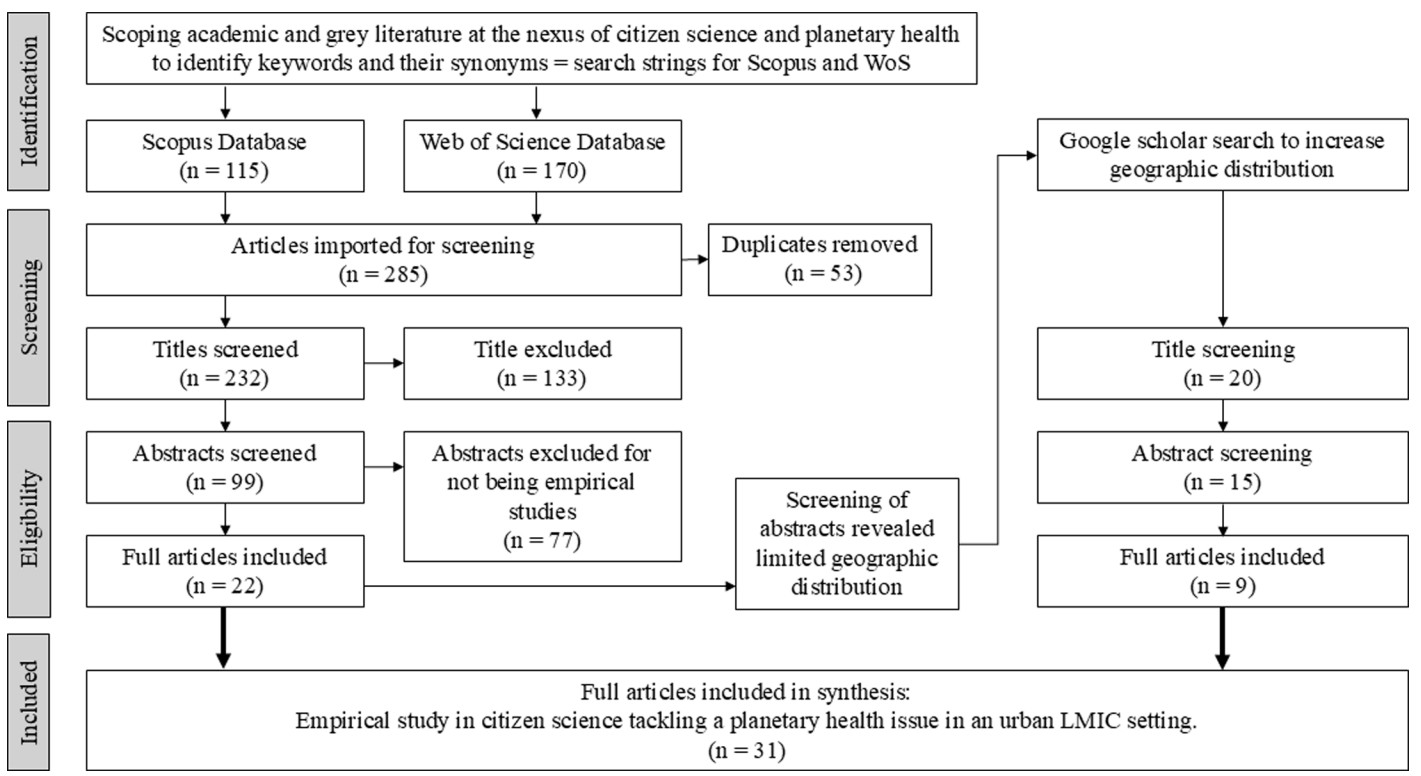

**Fig 3. Search and screening strategy (Source: Authors).**

**Table 1. Theoretical framework for the scoping review (Source: Authors).**

| Scoping review questions | Data extraction topics (adapted from JBI methodology) | |
|---|---|---|
| What are the methodologies used, and what human/planetary health outcomes are measured, in citizen science projects that focus on promoting planetary health in urban LMIC settings? | Background and Study Characteristics | • Research lead institution<br>• Funding source or agency<br>• Geographic location of the Study |
| | Methodologies | Citizen science<br>• Demographic factors<br>• Selection<br>• Consent<br>• Training<br>• Motivation |
| | | Data<br>• Collection tools and methods<br>• Analyses<br>• Validation |
| What are the lessons and challenges in the studies that could inform enhanced practices for initiating and sustaining citizen science projects to promote planetary health in LMICs? | Challenges and limitations | • Studies limitations<br>• Recommendations<br>• Future research |

lead institution of the study. In over half the studies, 20 out of 31 studies, the leading institutions were from the same geographical location as the study area suggesting that these 20 studies were led by institutions in the Global South. However, only five of these studies had funding sources from the Global South.

**Table 2.** Background characteristics of reviewed studies (Source: Authors).

| Study reference | Study l ocation | Lead i nstitu- tion country | Funding source | Study area(s) type | Researched topics |
|---|---|---|---|---|---|
| [25] | Argentina | Argentina | Argentina | Urban and Peri-urban areas | Biodiversity: Rumina decollata land snail |
| [26] | Argentina | Argentina | Argentina | | Biodiversity: *Iris pseudacorus* plant |
| [27] | China | China | China | Urban | Biodiversity: Woody plants monitoring |
| [28] | Powai Lake, Mumbai in India | India | India | Urban | Water quality and pollution |
| [29] | Jakarta, Bogor, Depok, Tangerang, Bekasi in Indonesia | Indonesia | Indonesia | Urban | Biodiversity: Butterfly species |
| [30] | Nairobi, Kenya | Kenya | UK | Urban | Air quality and pollution: capture pol- lutants measured - PM2.5 and PM10 |
| [31] | South Africa | South Africa | Not stated | | Biodiversity: Birding data |
| [32] | Cape Town, South Africa | South Africa | USA | Urban | Human health: Physical activity |
| [33] | Nigeria | Nigeria | Sweden and Switzerland | | Biodiversity: Birding data |
| [34] | Estación Biológica Corrientes, Argentina | Argentina | No funding | Peri-urban | Biodiversity: Howler monkeys and local conservation |
| [35] | Niger Delta, Nigeria | Nigeria | Netherlands and other sources | Urban and Peri-urban | Land use and biodiversity: Oil spill hazards and wildlife monitoring |
| [36] | Bogota in Colombia | Colombia | Not stated | Urban | Air quality and pollution: capture pollutants measured - PM2.5 |
| [37] | Mexico, Guatemala, El Salvador, Costa Rica, Panama, Colombia, Ecuador, Peru and Chile | Chile | EU and other sources | Urban | Waste pollution: Anthropogenic marine litter |
| [38] | Addis Ababa and other towns, Ethiopia | Ethiopia | UK | Urban | Flooding |
| [39] | Eastern region of Bolivia | Bolivia | Switzerland | Peri-urban | Biodiversity, water resources, and human activity |
| [40] | Rwanda | Rwanda | Germany | Rural and Urban | Human health: Cardiovascular dis- ease risk perception |
| [41] | Valle de Bravo basin, Mexico City, Mexico | Mexico | HSBC | Urban | Water quality and pollution |
| [42] | Akaki catchment, Awash River Basin, Ethiopia | Ethiopia | UK | Urban | Water sources: Rainfall |
| [43] | Xochimilco peri-urban wetland, Mexico City in Mexico | Mexico | HSBC | Peri-urban | Water quality and pollution |
| [44] | Malawi, Ethiopia and Rwanda, and urban South Africa | South Africa | Germany | Rural and Urban | Human health: Cardiovascular dis- ease risk perception |
| [45] | Sao Paulo, Brazil | United Kingdom | Multiple sources | Urban | Land use: socio-spatial dynamics |
| [46] | Luanda, Angola; Maputo, Mozambique | South Africa | UK | Urban | Land use: SDG 11 - Urbanisation and wellbeing |
| [47,48] | Argentina | France | France and other sources | Urban | Flooding |
| [49] | Lagos State, Nigeria | Canada | Canada and USA | Urban | Human health: Physical activity |
| [50] | Belo Horizonte, Brazil | Not stated | Multiple sources | Urban | Water quality and pollution |
| [51] | Lagoon Aghien, Abidjan | France | Not stated | Peri-urban | Water quality and pollution |
| [52] | Medjerda catchment in Tunisia | Belgium | Not stated | Peri-urban | Water sources: Rainfall |
| [53] | Mukuru informal settlement, Nairobi in Kenya | USA | Canada | Urban | Human health: Climate related health risks in informal settlement |
| [54] | Hamra, Beirut in Lebanon | United Kingdom | UK | Urban | Human health: Prosperity and quality of life |
| [55] | Ruhuha sector, Bugesera district in Rwanda | Netherlands | Netherlands | Peri-urban | Human health: Malaria vector surveillance |

*(Continued)*

**Table 2.** (Continued)

| Study reference | Study l ocation | Lead i nstitu- tion country | Funding source | Study area(s) type | Researched topics |
|---|---|---|---|---|---|
| [56] | Sunyani in Ghana | Germany | UK | Urban | Biodiversity, ecosystem services and urban green spaces |

## 2. Planetary health topics

We analysed the prevalent planetary health research topics investigated in the 31 studies. As shown in Fig 4, biodiversity and human health emerged as the most investigated topics. In biodiversity studies, the research either focused on a specific plant or animal species or broadly tackled related environmental and ecosystem issues. Human health studies also varied by focusing either on epidemiological factors or perceptions (Table 2). Water related studies were also common, focusing on water quality and pollution, water sources, and flooding. The studies focusing on land use investigated various indicators of socio-spatial dynamics in relation to biodiversity. Other topics identified were air quality and waste pollution.

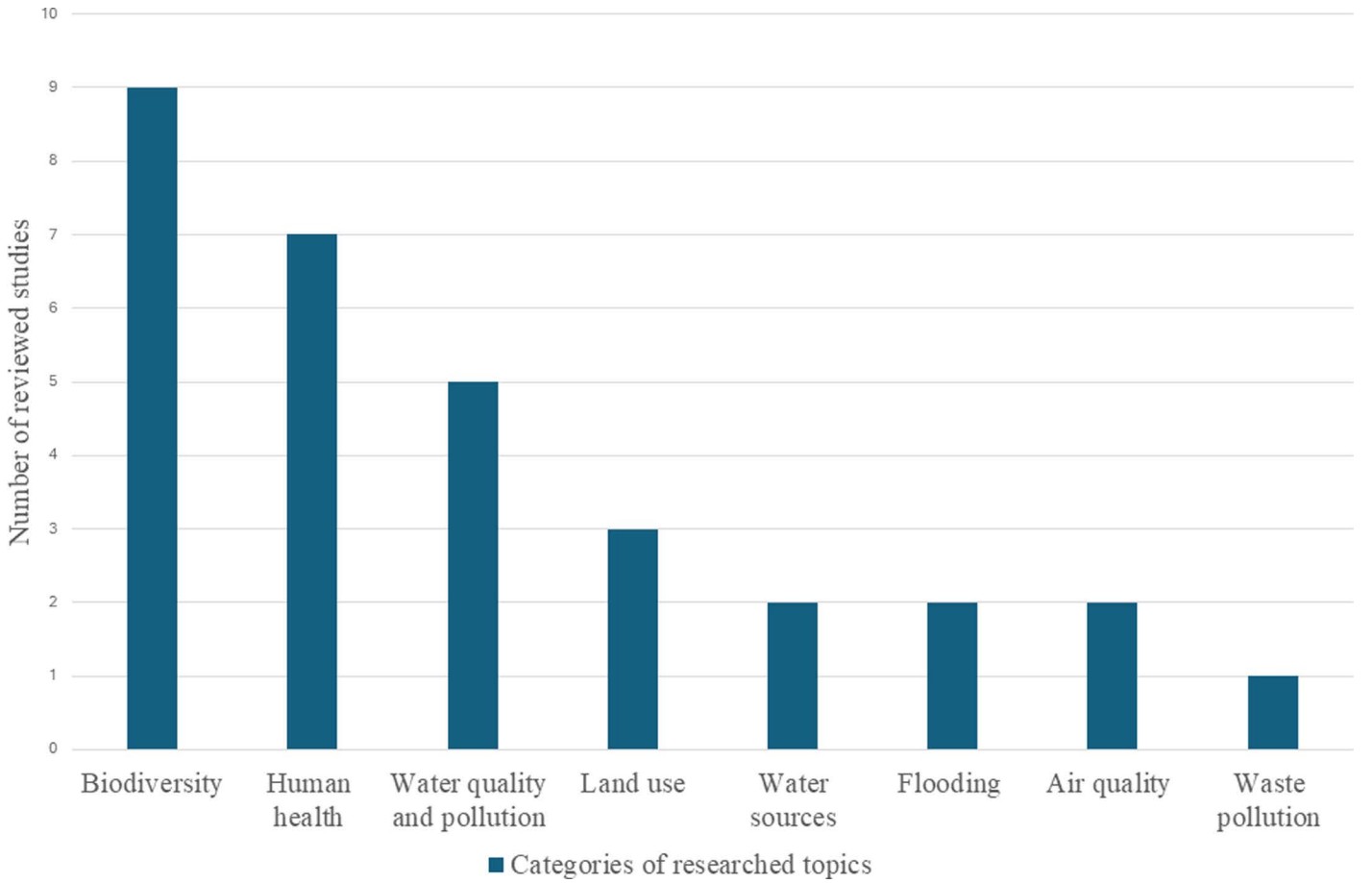

**Fig 4. Categories of researched topics on the reviewed studies (Source: Authors).**

**Table 3. Demographic information of citizen scientists in the reviewed studies (Source: Authors).**

| Study reference | Demographic information |
|---|---|
| [34] | High school students (from 13 to 17 years old) |
| [32] | Physically active participants (between 21–45 years, both genders) |
| [39] | 71% men and 29% women, age range was 25–60 years |
| [49] | Older persons aged 60 years or more |
| [56] | High school and university students, pupils, and non-students |
| [46] | Community members and university students |
| [35] | Members from affected communities (youths and students included) |
| [51] | Men, women, youth |
| [55] | Community members (men and women) |
| [42] | Men and women |
| [30] | Affected households |
| [38] | Community members |
| [52] | Citizens from different generations and educational backgrounds |
| [54] | Diverse community members |
| [47] | Local organisation members |
| [36] | Members of an activist group |
| [50] | Students |

## 3. Methodologies

We sought to gain a comprehensive understanding of how citizen scientists were engaged in research activities in LMIC urban settings. We first explored the demographic factors considered in the selection of volunteers and analysed the diverse methodologies employed in integrating them as citizen scientists. We then delved into the data approaches, investigating the variety of data tools and methods used in collecting, analysing, and validating data across the reviewed studies.

**3.1. Demographic factors in selecting citizen scientists.** Demographic information of the citizen scientists reported included age, sex, education level, and economic status. Seventeen of the 31 studies (55%) provided demographic information of the citizen scientists as shown in Table 3. Of the 17 studies that captured demographic data, only four studies (24%) stated the age range of the citizen scientist participants while six studies (35%) included or focused on children or youth.

**3.2. Selection of citizen scientists.** We explored how citizen scientists were selected, trained, compensated or motivated to participate in the studies. Recent studies show the importance of having well-defined recruitment and retention strategies especially in studies where long-term engagement is desirable [57,58]. To begin with, we analysed how citizen scientists were reached. Reaching out to local communities through community leaders or existing programs was the most common mode of finding volunteers, followed by reaching out to communities-of-practice such as schools, universities and non-governmental organisations [30,32,35,45] and/or reaching out to prospective volunteers from target groups such as community-based organisations or educational institutions [34,46]. Other studies reached out to targeted groups via emails and to the general public through social media [37]. Of note were studies where community leaders were consulted or engaged in selecting the volunteers. For instance, in the Mitroi et al study in Abidjan, local authorities and chiefdoms were consulted [51]. In another study in Rwanda, community outreach included various levels of government so as to include local and national government officials alongside local community members and experts in the study [40].

We noted that in some studies, citizen scientists were identified through longer-term engagements. For instance, Mintchev and co-workers described the citizen science engagement on quality of life in Beirut, Lebanon as an open-ended, three-year research engagement between university researchers and local communities in Beirut, resulting in 'interventions that address pressing challenges.' [54]. Similar results of implementing interventions from long-term citizen science work were reported by Corburn and coworkers. In their study of fieldwork led by a grassroots organisation known as *Muungano Alliance* in *Mukuru* informal settlement, they report on the significance of the integrated development plan and climate justice strategy for the settlement known as the '*Mukuru* Special Planning Area' that resulted from sustained engagements between the community and stakeholders from local universities and government [53]. These two examples from the reviewed studies illustrate the development benefits that sustained citizen science engagements can achieve. A third study drew from the ongoing work of an activist group known as the CanAirIO community in Bogota under which citizen scientists self-mobilise to monitor air pollution in Bogota [36].

In other studies where professional scientists sought to enhance large datasets, online outreach was the preferred mode of finding volunteers including through dedicated project websites or through social media or emails. For instance, in a study of the distribution of an invasive species of snail known as *Rumina decollata* across Argentina, the authors of the study recruited volunteers through the social media platform, Facebook resulting in 696 records of snail sightings in 179 cities across 16 provinces of Argentina. From the breadth of the collected data, the authors confirmed that the invasive snail had spread rapidly in the country over the last three decades [25]. Similarly, another study from Argentina on an invasive plant known as *Iris pseudacorus*, sought to update the invasion risk maps by integrating citizen science reports into a wider study on the plant. The wider study included other conventional data collection methods using geographic information systems. By recruiting citizen scientists through Facebook and Instagram social media platforms, the authors reported receiving a larger and more diversified amount of data from citizen scientists as compared to their own conventional methods (ibid). A third study used the project website to gather data on the distribution of a species of butterfly across cities and towns in Indonesia [29].

Two studies based on the *Birdlasser app* demonstrated that the app enhanced the participation of citizen scientists in the Southern African Bird Atlas Project [31] and in Nigerian Bird Atlas Project [33]. As emphasised in the South African study [31], submission by citizen scientists per year increased by 155% when the *Birdlasser app* was introduced. Prior to the app, recruitment was laborious as professional scientists had to send instruction booklets and printed checklists to prospective volunteers, few of whom responded to the call.

**3.3. Seeking consent from citizen scientists.** Consent is crucial in citizen science initiatives to ensure that participants understand the purpose of the study and agree to participate voluntarily. In the reviewed studies, few detailed how consent was obtained from citizen scientists. In three studies, volunteers who attended the inception workshops were required to consent to participating in the study [30,32,49]. In a study with school children, head teachers and parents or guardians gave consent for the children to participate [56]. In another study involving students in designing a bridge as a local conservation intervention, the students themselves obtained the necessary permissions from the local authorities to install the bridge [34]. Other studies detailed the institutional permissions that were sought to carry out the study including permission from local authorities [51] approvals by regulatory bodies [26,40] and ethical clearance from academic institutions [37,46].

In a study on environmental issues in the Niger Delta, the authors report on incidents where citizen scientists who were monitoring oil spills were harassed by security agents guarding the oil spill facilities or were not allowed to engage certain stakeholders. These challenges were mitigated by advocating for mainstreaming of local monitoring as a strategic participatory management process [35]. This case highlights the importance of informed consent by volunteers who may be putting themselves at risk by becoming citizen scientists.

**3.4. Training and engaging citizen scientists.** The most common mode of training citizen scientists was through training meetings organised by professional scientists to discuss the study aims with citizen scientists and demonstrate

how to use the research tools. Twenty-three of the studies involved some form of in-person workshop or online training run by professional scientists. The level of direct engagement varied in the studies in terms of training intensity. For instance, in the study on air quality monitoring with informal settlement households in Nairobi, Kenya, professional scientists held a community meeting to demonstrate how to use the air quality sensors. However, the sensors were mounted on the rooftops of the selected households suggesting that the citizen scientists had a minimal role to play in the collection of data [30]. As such, the training appears to have been more consultative rather that empowering the households to carry out the study themselves.

Other studies demonstrated a higher level of citizen participation such as in the self-mobilisation efforts of citizen scientists through an activist group [36] or a grassroots group [53]. Two studies reported on the training of trainers indicating that professional scientists may have sought to delegate power to selected citizen scientists to play bigger roles in inducting other volunteers [40,41].

In other studies, professional scientists did not engage directly with citizen scientists as study guidelines and protocols were provided via a dedicated app or website, through social media, or through printed materials. In such cases the provided instructions had to be clear and easy to follow and even include pictorial guides [25,29]. In the case of the *Birdlasser app*, citizen scientists had easy to follow prompts and pictorial guides that facilitated the data collection process [31,33]. In these cases, the citizen scientists had full control over the data collection process.

**3.5. Participation and engagement of citizen scientists.** Arnstein's ladder of citizen participation describes different degrees of engagement ranging from low (non-participation) to medium (degree of tokenism) to high (degree of citizen control) [59], with potential for engagement of citizen scientists across all aspects of research from design of the objectives/scope and design of data collection tools to participation in data collection, analysis and interpretation of findings. One study displayed a high degree of citizen control as self-selecting grassroots activists from the CanAirIO community in Bogota engaged in all aspects of the research including defining their research agenda, building their own low-cost air quality monitors, collecting and analysing air quality data. The activists then used the data to advocate for improved environmental governance and change in air quality standards [36]. Two other studies reported wide consultation of stakeholders in refining the research agenda [40,46] but did not elaborate on how citizen scientists themselves were included in designing the research tools.

In most of the studies, citizen scientists collected the data and submitted it to professional scientists, displaying differing levels of engagement ranging from merely collecting data to advocating for change. More specifically, in studies that required specialised technical expertise or collected large quantitative datasets, citizen scientists participated only in the collection of data [25,27,30,31,33,41,43,47,50,52]. In other more qualitative studies, citizen scientists were involved beyond data collection. For instance, in two studies on physical activity using the *Stanford Neighbourhood Discovery tool*, citizen scientists collected data around their community and then participated in the thematic analyses of findings in workshops with the professional scientists [32,49]. The *Discovery tool* is part of the *Our Voice* approach by Stanford University that provides for the participation of citizen scientists in a four step process of 'Discover - Discuss – Activate – Change'. According to King and co-workers, this four-step process empowers residents to initiate health-promoting change in their neighbourhoods in collaboration with stakeholders [60]. In another study on alternative urban futures in Sao Paulo in Brazil, citizen scientists were engaged as community mappers who collected data on community walks and participated in conversation circles to discuss the findings with professional scientists and other stakeholders [45]. In another study on developing conservation interventions for monkeys, citizen scientists including students and teachers worked with professional scientists to design interventions in the form of canopy bridges, one of which was later built [34].

**3.6. Motivating citizen scientists.** Another emerging issue in engaging citizen scientists was how to motivate and compensate citizen scientists. In four studies, citizen scientists were given monetary compensation for their participation [30,32,42,54]. Manshur and co-workers acknowledged that the monetary compensation may have introduced biased participation into the study. For Okop and co-workers, financial compensation was not deemed necessary since the

citizen scientists would be collecting data from their neighbours who would readily participate in the study. Nevertheless, the citizen scientists were given a stipend for transport and meals for the days when they engaged in the fieldwork [44]. Similarly, Zabbey and co-workers reported that citizen scientists were provided with transport, snacks and water during fieldwork [35].

In one study, citizen scientists were recognised for their submissions with certificates [27], while in another, the top submitters of data were recognised and gifted with branded merchandise from one of the stakeholders [35]. Another study sent appreciation text messages to data contributors [51].

Promoting teamwork and a sense of community was another way to motivate citizen scientists by pairing them with professional scientists during their fieldwork. In this regard, Tende and co-workers reported that unskilled bird enthusiasts were paired with formally trained ornithologists to motivate them and promote skills transfer, teamwork, and data quality control [33]. Other forms of motivation for citizen scientists could be in seeing their proposals implemented to solve their problems such as in the case of informal settlement dwellers who generated evidence that led to the climate action strategy for their settlement [53], or students whose prototype for a bridge for local conservation was built [34].

Engaging with decision-makers is one way to increase chances for implementing proposals emerging from citizen science initiatives. In a few of the reviewed studies, policy advocacy was highlighted as an outcome of the citizen science engagement with local or national government officials. For instance, the self-selecting citizen scientists monitoring air quality in Bogota, were able to instigate public discussions on environmental governance and influence change in air quality standards for their city [36]. These examples illustrate the diversity in how citizen scientists engage in the research process, ranging from merely collecting data to advocating for change.

**3.7. Data tools and methods.** The data collection tools used by the citizen scientists in the reviewed studies were diverse as detailed in S4 Table. We identified five broad categories of data handling approaches utilised in the citizen science projects, summarised in Table 4.

Notably, 12 studies used mobile applications indicating the prevalence of smartphones in the urban contexts of the studies. Notably, the commonly used mobile applications were *Birdlasser* app [31,33] *Stanford Discovery Tool* app [32,49] *KoboToolBox* [29,46] *Epicollect5* [40,51]. These apps are free to use with *KoboToolBox* and *Epicollect5* being open source.

**3.8. Data validation and verification.** Data from non-professional volunteers engaging in citizen science initiatives often face concerns regarding quality and validity due to variations in experience, training, and the methodologies used in the initiatives. Consequently, professional scientists who engage citizen scientists have to find ways to validate the submitted data and ensure accuracy and reliability [58]. Using insights from citizen science work done in India, Vattakaven and co-workers propose data validation methods such as designing high quality training protocols, calibrating equipment, as well as using expert verification, cross-validation, automated quality checks, comparison and replication (ibid).

These proposals for data validation methods are evident to varying degrees in most of the reviewed studies. Protocol checks were used in the malaria vector study to confirm that correct procedures were followed in collecting samples [55]. Calibration of equipment was described in detail in the air quality study by Manshur and co-workers. The Authors detail how the optical particle sensors were calibrated pre-sampling using gravimetric measurements carried out in a laboratory. From this pre-calibration, the authors reported that out of 15 sensors, only one recorded inaccurate measurements and was thus excluded in the analyses [30].

Cross-checking and expert verification were other methods used to validate citizen science submissions against conventional measurements. For instance, in the study of snail distribution by Piza et al, geographic coordinates submitted by citizen scientists were cross-checked using Google Maps and the point-radius method to estimate the certainty or uncertainty of location accuracy. In air quality monitoring, professional scientists downloaded data from sensors and measured the particulate matter [30]. In water quality and pollution monitoring, samples were sent for laboratory analyses [41,51] or water related measurements were analysed using specialised analytical tools or software [43,50,52]. Similarly, distribution data and geographic data were analysed using specialised software [25,27,31,47].

**Table 4.** Categories of citizen science data tools form the reviewed studies (Source: Authors).

| Categories | Specifications from the reviewed studies |
|---|---|
| Proprietary equipment | • *AlphaSense OPC-NC* sensors and *CanAirIO* low-cost devises for air quality monitoring.<br>• *Alfakit colorimetric kits* and *Alabama Water Watch LaMotte Kit 21* for water quality measurements.<br>• *Actical accelerometers version B-1* for measuring physical activity. |
| Non-specialised equipment and DIY tools | • Smartphones, cameras, and tablets for recording images and maps.<br>• Hand augers for soil sampling; hand nets for capturing macroinvertebrates.<br>• Ropes, sticks, and measuring tape for measuring rainfall run-off.<br>• Handmade carbon-dioxide baited traps to capture mosquitoes. |
| Mobile applications | • *KoboToolBox app*<br>• *Epicollect5 app*<br>• *BirdLasser app*<br>• *Utrees app*<br>• *ArcGIS Survey123*<br>• *WhatsApp* |
| Aggregator web platforms | • *BirdLasser.com*<br>• *KupuKita.org*<br>• *iNuralist.org*<br>• *Google forms* |
| Paper-based forms | • Questionnaires<br>• Data sheets |

Automated quality checks were also possible in cases where apps or web platforms were used as evidenced in the woody plants study that utilised the *Utrees app*. The app had an added function of automatic species identification [27]. Similarly, the *BirdLasser app* used in two of the studies has an automated identification function that analyses citizen scientists' photographs against a database of known species. In one of these studies a second layer of data validation involved expert verification of geographic locations of the species sightings. In this regard, an entry was flagged as out of range if it was sighted outside of its known range and/or season of the year. Submitters of flagged entries were contacted for clarification before acceptance into the database [33].

Similarly, photographs of requested species in another study were checked by professional scientists who discarded erroneous or doubtful reports [26]. Expert verification and cross-checking with previous empirical data was also used by Rano and co-workers by involving architects and engineers in the co-design process with students [34]. Stakeholder engagement and pairing citizen scientists with professional scientists was another method of expert verification in several studies [35,40,46,49]. Comparing citizen science data with expert data was also used as a validation method in two studies [50,51].

On data accuracy, two studies reported on the percentage of validity of citizen science submissions. These included the study on anthropogenic marine litter that reported 74% accuracy of submissions after expert review [37], and the study on environmental monitoring where satellite imagery was used to confirm 84% accuracy of citizen science submissions [39]. In two other studies, citizen science data was compared to reference data confirming that the referenced sets yielded excellent agreements [42,52].

## 4. Challenges and limitations of citizen science

The reviewed studies reported varied challenges and limitations including:

• Inconsistencies in referenced data: Issues with spatial data accuracy were reported, impacting the reliability of geographic analyses [29].

• Lack of demographic data: The absence of demographic information hindered understanding of citizen scientists' skills and backgrounds, which could have enhanced the research outcomes [27].

- Internet connectivity issues: Problems with slow or missing data submissions were attributed to poor internet connectivity thus affecting the completeness and timeliness of data [32,39].

- Limited study periods: Limited study durations failed to capture seasonal variations in air quality, impacting the comprehensiveness of monitoring [30].

- Barriers to citizen participation: Unrest in regions such as Northern Nigeria led to low citizen engagement and participation [33].

- Uneven participation from local government officials: Participation of local officials was uneven and influenced by external pressures, such as upcoming local elections, which affected their involvement [46].

- Harassment of citizen scientists: Fieldwork in certain areas involved harassment, which undermined the safety and effectiveness of citizen scientists [35].

- Low-quality submissions: Issues with the quality of data submissions were observed, impacting the overall reliability of the research findings [47,51].

- Sustaining long-term engagement: Challenges in maintaining ongoing engagement between citizen scientists and stakeholders were noted, affecting the continuity and impact of projects [40,52,53].

- Generalisability of findings: Concerns were raised about the generalisability of findings from localised studies to broader contexts [53].

## Discussion

In this scoping review, we explored the use of citizen science approaches and tools to investigate planetary health in urban and urbanising LMICs.

## Funding and lead institutions

Studies have shown that research funding often follows patterns of global inequality thus favouring Global North institutions [60]. Additionally, as global science remains primarily Anglo-American, non-English and indigenous knowledge systems continue to be peripheral. We acknowledge that this paper's exclusion of non-English articles perpetuates this hegemony [61]. We acknowledge that this paper's exclusion of non-English articles perpetuates the hegemony of Anglo-American studies thus excluding other non-English knowledge systems.

The findings indicated that citizen science research in LMICs continues to be predominantly driven by institutions and funding agencies from the Global North which may reinforce existing power imbalances in global health research [62]. These findings may also indicate that citizen science projects in LMICs have yet to attract substantial local funding, leadership or ownership. To further understand the dynamics at play, a comprehensive meta-review of citizen science studies in the Global South is required. Such a review could elucidate emerging patterns in funding flows, stakeholder engagement, and research agenda-setting, and ultimately inform strategies for more equitable and locally led citizen science initiatives in LMICs.

Additional research is also needed to understand the complexities of institutional collaborations across the Global North and South. For instance, in the study by Corburn et al [53], the authors report on a citizen science initiative carried out in Nairobi, Kenya by a grassroots organisation known as *Muungano Alliance*. In the report, the corresponding author's institution is the University of California in the United States of America (USA) hence we categorised the USA as the lead institution country. However, we take note that the other three co-authors of the report are from the *Akiba Mashinani Trust* which manages the *Muungano Alliance*. Therefore, while we categorised the USA as the lead institution country, such an approach requires further investigation, which is beyond the scope of this review.

Policy and regulatory instruments can further enhance leadership of Global South institutions in citizen science projects by institutionalizing frameworks that promote open and citizen science. For example, Argentina's Ministry of Science, Technology, and Innovation has implemented a law that mandates open access to publicly funded research. Within this framework and through its National Citizen Science Program, Argentina is promoting citizen science initiatives, funding, and resources [63]. Moreover, addressing structural challenges like the digital divide through national infrastructure, as exemplified by Argentina's hybrid cloud initiative, would enable local institutions to participate more effectively in global research efforts. Additionally, the promotion of multilingualism as done in Argentina would ensure broader societal engagement in citizen science (ibid). Such policies and frameworks would create a virtuous cycle where citizen science benefits from open science initiatives, and vice versa, which would potentially position Global South institutions as emerging leaders in citizen science research.

## Researched topics

The diversity of researched topics from the findings further displays the multiple intersecting points between citizen science and planetary health. Our findings highlight a predominant focus on biodiversity and human health, illustrating the potential for greater use of citizen science to explore other planetary health topics highly relevant to urban LMICs such as flooding, extreme weather and air pollution. More studies in this regard are needed in LMICs given that they face the most adverse effects of global environmental changes. In addition, while only one reviewed study made reference to the context of regional unrest [33], the prevalence of political instability in many LMIC settings suggests the need for a better reporting and understanding of the practical experience of conducting citizen science research under such conditions.

## Potential for technology

While paper-based forms remained an essential method in areas with limited digital access, allowing participation in data collection in resource-constrained settings, we noted the potential of technology to enhance citizen science projects. Such technology included the use of proprietary equipment, such as custom sensors and tools designed for specific research purposes, and non-specialised equipment, like smartphones or do-it-yourself (DIY) tools, which enabled widespread participation without requiring significant investment in technology.

Digital tools, such as dedicated apps, were also found to enhance participation by offering clear, user-friendly instructions, empowering volunteers to take more control over data collection and submission. Such mobile applications also enabled real-time monitoring and instant feedback to participants enhancing engagement and data accuracy. Aggregator web platforms facilitated the integration and visualisation of data collected from diverse sources, supporting large-scale analyses and collaborations across multiple regions.

Despite their usefulness, technologies in the reviewed studies presented limitations such as data quality due to variations in experience, training, and the complexity of methodologies used in the initiatives. Consequently, professional scientists found ways to validate citizen science data and ensure accuracy and reliability. Professional scientists also enhanced data validity through protocol checks, equipment calibration, and expert verification of data. Automated quality checks were also possible in some apps or web platforms. Stakeholder engagement and pairing citizen scientists with professional scientists was another method of expert verification in several studies.

Technologies such as mobile apps and web platforms may exacerbate the digital divide by excluding participants who have poor internet access and lack the technical capabilities [32,39]. Data security, transparency, and autonomy are also issues of concern where open data platforms are used. These considerations underscore the need for inclusive strategies to ensure equitable participation when technologies are used in citizen science initiatives.

## Standardising reporting of citizen science planetary health studies

We explored different approaches to designing and implementing planetary health citizen science studies from recruitment, consent and motivation strategies to training, approaches to data collection and degrees of involvement at different stages of the study process.

The level of engagement of citizen scientists varies widely across studies. Some involve participants only minimally, in tasks such as basic data collection, while others actively include them in data analysis and decision-making processes. Our findings suggested that quantitative data, derived from methods such as sampling, specimen counting, sensing, and measuring, which required laboratory settings or specialised analytical techniques, were less likely to report direct involvement of citizen scientists in analysis. In contrast, qualitative data from surveys, interviews, focus groups, and mapping which were analysed thematically, provided more opportunities for citizen scientists to actively participate in the analytical process.

The reviewed studies also highlighted several challenges and limitations in citizen science initiatives that impacted participation, data quality, and the overall effectiveness of the projects. Issues such as inconsistencies in data accuracy and the lack of demographic information about participants impacted the reliability and inclusiveness of research findings. Technical problems included poor internet connectivity and limited study periods. Additionally, external factors like regional unrest, harassment, and uneven participation from local leaders created barriers to sustained citizen engagement. Furthermore, challenges in maintaining long-term engagement between citizen scientists and stakeholders, as well as concerns about the generalizability of findings from localised studies, were identified as obstacles that limited the broader impact and applicability of these initiatives. These challenges and limitations underscore the need for improvements in methodology and support to enhance the effectiveness and reach of citizen science projects in LMICs.

We noted wide variability in the extent to which these data were reported across the reviewed studies. Based on these findings, we recommend a standardised reporting format (Table 5) for citizen science projects to facilitate comparison of projects:

## Towards guidelines for citizen science data reliability

The findings on data validation emphasised the significant role of gathering and using reliable citizen science to advance planetary health in LMICs. To improve the reliability and credibility of citizen science data, we propose the following recommendations (Table 6) from the findings to guide the creation of robust validation and reliability guidelines:

## Conclusion and recommendations

This scoping review sought to improve understanding of the extent to which citizen science approaches are being used to investigate planetary health in the Global South.

The majority of the studies identified focused on biodiversity, illustrating the as yet unharnessed potential of deploying citizen science to better understand evolving planetary health risks in Majority World cities.

Our findings highlighted significant avenues for future research to enhance the efficacy and equity of citizen science in urban settings of LMICs. These include a focus on reducing funding disparities, enhancing demographic representation, and developing robust recruitment and retention strategies, as well as data validation protocols to maximise impact.

Noting wide variability in the reporting of citizen science methods, we propose an approach to standardise the reporting to improve comparability. Informed by the review findings, we further suggest guidelines to improve data reliability, addressing an oft expressed concern with citizen science data.

We have observed in the reviewed studies that Global South institutions and governments need to enhance policy instruments and regulatory frameworks that will promote citizen science in LMICs cities. Other sources highlight that Argentina exemplifies the promotion of citizen science with its regulatory framework for promoting open and citizen science [63]. In India, stakeholders urge that the country needs to enhance policy responses that will promote a more

**Table 5. Proposed standardised reporting of planetary health citizen science projects (Source: Authors).**

| Reporting indicators | Illustrative categories |
|---|---|
| Planetary health topics covered | Environmental changes of planetary health significance such as:<br>• Flora and fauna biodiversity and natural ecosystems<br>• Water quality and pollution<br>• Land use<br>• Agriculture and forestry<br>Direct health effects: Green blue and grey infrastructure features that increase or decrease the risk of/ exposure to:<br>• Flooding<br>• Air pollution<br>• Heat<br>• Extreme weather<br>• Waste pollution<br>• Water pollution/ Ocean acidification<br>• Injury<br>• Vector-incubating environments<br>Human health behaviours and experiences that may be impacted by environmental changes such as:<br>• Physical activity<br>• Diet<br>• Migration/Displacement<br>• Aesthetics and culture<br>Human health outcomes that cascade from risk exposures and changes to health behaviours such as:<br>• Mental health, e.g., self-reported mood<br>• Physical health, e.g., cardiovascular risk |
| Selection and recruitment | • Community outreach to engage affected local communities or communities of practice<br>• Engagement of the wider public via social and legacy media |
| Demographics of citizen scientists | • Age<br>• Sex or Gender<br>• Target groups, e.g., sectoral interest groups |
| Consent | • Volunteer informed consent<br>• Institutional permissions |
| Training | • Professional scientists engage directly with citizen scientists through meetings and workshops to inform, discuss, and demonstrate methods.<br>• Professional scientists remotely provide guidelines on how to carry out study<br>• Use of digital tools such as mobile application to deliver or support training |
| Data collection tools | • Paper based data collection tools<br>• Technology-enabled data collection tools<br>• Non-specialised equipment<br>• Proprietary equipment<br>• Aggregator web platforms |
| Data handling | • Citizen scientists collect and submit data to professional scientists<br>• Professional scientists include citizen scientists in analysing and reporting on data |
| Motivation | • Monetary compensation<br>• Fieldwork resourcing – transport and meals<br>• Recognition and awards for good performance<br>• Implementation of citizen science proposals<br>• Policy advocacy |
| Level of engagement | Using frameworks like Arnstein's ladder of participation to describe the degree of engagement of citizen scientists such as:<br>• Informing<br>• Consulting<br>• Placation<br>• Partnership<br>• Delegation<br>• Full Control |

*(Continued)*

**Table 5.** (Continued)

| Reporting indicators | Illustrative categories |
|---|---|
| Stages of research with citizen scientist involvement | • Study design<br>• Design of data collection<br>• Data analysis<br>• Data interpretation<br>• Write up<br>• Engagement of decision makers/ implementation actors with findings and recommendations<br>• Design and evaluation of interventions |
| Challenges and Limitations | • Participation and inclusion<br>• Retention<br>• Data quality<br>• Implementation barriers and facilitators<br>• Contextual barriers (e.g., tensions and conflict, internet connectivity) and enablers (e.g., community leadership buy in; strategic partnerships) |

**Table 6. Proposed guidelines to improve reliability of citizen science data (Source: Authors).**

| |
|---|
| Protocol checks: Ensure that correct procedures are followed especially in cases of sample collection where accuracy is paramount. |
| Equipment calibration: Calibrate equipment before data collection using precise methods to identify inaccurate measurements after data collection. |
| Cross-checking and expert verification: Validate citizen science submissions by cross-checking data with conventional measurements and using expert verification to confirm accuracy. |
| Automated quality checks: Use apps or web platforms with automated functions for data validation, such as species identification, and incorporate additional expert review when needed. |
| Expert verification: Engage experts to review and validate data such as species identifications and geographic data. |
| Stakeholder engagement: Pair citizen scientists with professionals for additional oversight and validation and include them in decision-making processes. |
| Comparison with expert data: Compare citizen science data with expert data to ensure accuracy and reliability. |
| Data accuracy reporting: Report the accuracy of submissions by comparing them with reference data or using other verification methods. |

coherent citizen science approach [64]. The African Union is also leveraging Agenda 2063 to promote a people-driven development that will integrate citizen science into national and regional development plans [65]. More research is needed to articulate the impact of these policies and regulations in promoting citizen science in the Global South.

Another area for future research is a contextual analysis of citizen science projects to provide deeper insights into regional variations and qualitative dimensions such as power dynamics and knowledge systems. Such a qualitative analysis could further explore the nuances of community engagement and knowledge co-production, especially in cases that include highly marginalized groups. This contextualisation would offer valuable lessons for designing more inclusive citizen science initiatives.

Lastly, a notable finding was a paucity of information on the (extent of) involvement of youth in studies identified. Given the rapidly growing youthful demographic in cities within LMICs, there is a critical need for further research into how youth-driven citizen science initiatives can enhance and accelerate advancements in planetary health, both through contributions to research and through evidence-informed advocacy. Understanding how young people engage with and contribute to these projects could unlock significant potential for innovation and mobilisation in addressing planetary health while also mitigating the constraints of growing citizen science work in the Global South.

## Supporting information

**S1 Table. PRISMA ScR checklist.**
(DOCX)

**S2 Table. Search strategy and search strings.**
(DOCX)

**S3 Table. Data extraction template.**
(DOCX)

**S4 Table. Details of the data handling methods and tools from the reviewed studies.**
(DOCX)

## Acknowledgments

We acknowledge the contribution of Dr Monika Kamkuemah in the early stages of planning this review.

## Author contributions

**Conceptualization:** Tolu Oni.

**Data curation:** Amollo Ambole, Christer Anditi, Tolu Oni.

**Formal analysis:** Amollo Ambole, Christer Anditi, Tolu Oni.

**Funding acquisition:** Tolu Oni.

**Investigation:** Tolu Oni.

**Methodology:** Amollo Ambole, Tolu Oni.

**Project administration:** Tolu Oni.

**Supervision:** Tolu Oni.

**Validation:** Amollo Ambole, Christer Anditi, Tolu Oni.

**Visualization:** Amollo Ambole.

**Writing – original draft:** Amollo Ambole.

**Writing – review & editing:** Tolu Oni.

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
