## [Decision Letter · Decision Letter 0]

3 Jan 2025

PGPH-D-24-02515

Potential of citizen science to advance urban planetary health research in low and middle-income countries: A scoping review

Dear Dr. Tolu Oni,

Thank you for submitting your manuscript to PLOS Global Public Health. After careful consideration, we feel that it has merit but does not fully meet PLOS Global Public Health’s publication criteria as it currently stands. Therefore, we invite you to submit a revised version of the manuscript that addresses the points raised during the review process.

We look forward to receiving your revised manuscript.

Kind regards,

Muhammad Asaduzzaman, MD MPH MPhil

Academic Editor

Journal Requirements:

Additional Editor Comments (if provided):

Reviewers' comments:

Reviewer's Responses to Questions

**Comments to the Author**

1. Does this manuscript meet PLOS Global Public Health’s publication criteria ? Is the manuscript technically sound, and do the data support the conclusions? The manuscript must describe methodologically and ethically rigorous research with conclusions that are appropriately drawn based on the data presented.

Reviewer #1: Yes

Reviewer #2: Yes

Reviewer #3: Partly

Reviewer #4: Yes

2. Has the statistical analysis been performed appropriately and rigorously?

Reviewer #1: N/A

Reviewer #2: Yes

Reviewer #3: Yes

Reviewer #4: N/A

3. Have the authors made all data underlying the findings in their manuscript fully available (please refer to the Data Availability Statement at the start of the manuscript PDF file)?

Reviewer #1: Yes

Reviewer #2: Yes

Reviewer #3: Yes

Reviewer #4: Yes

4. Is the manuscript presented in an intelligible fashion and written in standard English?

Reviewer #1: Yes

Reviewer #2: Yes

Reviewer #3: Yes

Reviewer #4: Yes

5. Review Comments to the Author

Reviewer #1: It is my pleasure to review this manuscript which is highly relevant in the context of planetary crisis. It will well written. However, the findings and discussion sections are mixed up and difficult to differentiate. I suggest to carefully revisit findings and discussion section for easy understanding to readers.

Reviewer #2: Peer Review Report: Potential of Citizen Science to Advance Urban Planetary Health Research in Low and Middle-Income Countries: A Scoping Review

1. Overall Impression:

This scoping review makes a valuable contribution to the field by systematically exploring the use of citizen science in urban planetary health research within low and middle-income countries (LMICs). The paper effectively highlights the underutilized potential of citizen science in this context and identifies key methodological considerations and challenges. The call for standardized reporting and a focus on equitable funding is particularly timely and relevant. While the review is well-structured and written, some areas require further development to enhance its impact and rigor.

2. Strengths:

• Clear Research Question and Scope: The research question is clearly defined, and the inclusion/exclusion criteria are well-articulated, providing a focused and manageable scope for the scoping review.

• Systematic Approach: The authors followed established methodologies (PRISMA-ScR and JBI) for conducting a scoping review, enhancing the rigor and transparency of the process.

• Relevant and Timely Topic: The topic is highly relevant given the growing interest in planetary health and the potential of citizen science to address global health challenges, particularly in LMICs.

• Critical Analysis of Funding: The analysis of funding sources and the emphasis on the need for greater diversity and equitable funding distribution are crucial observations that contribute significantly to the paper's policy relevance.

• Well-Structured and Written: The paper is well-organized, easy to follow, and written, making it accessible to a broad audience.

• Actionable Recommendations: The authors provide concrete recommendations for future research and for standardizing reporting practices in citizen science planetary health projects.

3. Areas for Improvement:

• Depth of Qualitative Analysis: While the review identifies challenges and limitations, a deeper qualitative analysis of the included studies could provide richer insights into the contextual factors influencing the success or failure of citizen science initiatives. Consider exploring themes related to power dynamics, community engagement strategies, and the role of local knowledge systems.

• Geographical Specificity: While the review focuses on LMICs, further analysis of regional variations in the application and effectiveness of citizen science would strengthen the findings. Are there significant differences between regions or types of urban settings?

• Discussion of Technology: The role of technology in citizen science is mentioned, but a more in-depth discussion of the types of technologies used, their effectiveness, and their limitations (e.g., digital divide, data security) would be beneficial.

• Integration of Grey Literature: The review primarily focuses on peer-reviewed literature. Including relevant grey literature (e.g., government reports, policy documents, NGO reports) could provide a more comprehensive picture of the EIDM landscape.

• Strengthening the Conclusion: The conclusion could be strengthened by more explicitly linking the findings to specific policy recommendations and by outlining potential pathways for implementing these recommendations.

4. Policy Implications:

The findings of this scoping review have significant policy implications, particularly regarding funding allocation for citizen science initiatives in LMICs. The emphasis on the need for a greater diversity of funding sources and a shift in the center of gravity of funding decisions to better align with contextual realities in the Global South is a crucial policy message. The recommendations for standardized reporting will also improve the comparability and usability of citizen science data for policymaking.

5. Overall Recommendations for Revision:

• Expand Qualitative Analysis: Incorporate a more in-depth qualitative analysis of the included studies to provide richer insights into contextual factors.

• Address Geographical Variations: Analyse regional variations in the application and effectiveness of citizen science.

• Deepen Technological Analysis: Provide a more detailed discussion of the role of technology in citizen science initiatives.

• Incorporate Grey Literature: Include relevant grey literature to provide a more comprehensive overview.

• Strengthen Conclusion and Policy Recommendations: More explicitly link the findings to specific policy recommendations and outline potential pathways for implementation.

• Consider Visualizations: To improve readability and impact, enhance the presentation of data with clear and informative visualizations (e.g., maps, charts, and diagrams).

By addressing these points, the authors can significantly strengthen the manuscript and enhance its contribution to the field of citizen science and planetary health research. The paper has the potential to be a highly influential contribution to the literature.

Reviewer #3: Thank you for the choosing an interesting and timely topic. Although you selected significant features of citizen science in the articles, but did not discussed clearly how much participatory those studies were. Citizen science is ideal to reverse the gaze towards community's perspectives but the projects using citizen sciences often adopt same top-down approach for selecting the design and participants. You may like to reflect more on this bias in your analysis.

I would also like to see your reflection on challenges of these projects faced for engaging citizens and how they overcome participation bias.

Reviewer #4: Overall: This study fills a gap in the literature and presents a very detailed methodology for the scoping review. A reader will learn more about the citizen science and planetary health nexus in contexts that are the most vulnerable to global environmental change. The article does an excellent job at presenting practical applications and a framework/guidance to support a certain degree of consilience for future research to consider. The body of work, therefore, is necessary to strengthen planetary health citizen science research. The supporting information provides clarity and will ensure continuity in future research. Thank you for your submission.

---

Line 51: “Recent assessments…” Can you provide a citation for this? If it is (4), I would also suggest indicating that there.

Line 59-63: Should the (6) definition be block text?

Line 66: I believe the authors mean ‘Figure 1’ here? Figure 2 appears later in the manuscript, while Figure 1 appears first.

Line 146: This is where the other figure error is. Please revise. The diagram is clear in steps. It is a bit blurry, but I assume that is how the submission usually appears on a manuscript submission.

Line 67: Would examples of the escalation of human pressure in parentheses be of help here? Also, when I look at the figure, I am left wanting details of what those are. Examples of them somewhere, either the body, or in the figure, would help.

Lines 22 & 73: Confirming that “Environmental change” is the intended phrase here, and not “global environmental change.”

Line 144: I am wondering if all three of the reviewers initially reviewed separately and then came together to collaboratively come to a shared agreement? The iterative team approach could be described in more detail here.

Line 151: Search and screening strategy figure title could use more detail. For example, “...strategy for conducting a….”. Also, can the “Screening of abstracts revealed limited geographic distribution” be connected to an additional or different step? At the moment it reads:

99 abstracts were screened, 77 were excluded, and from those that were excluded, their abstracts were screened again and revealed that there was a limitation in geographical distribution (which is surely not the case here). Do the authors mean that they conducted an additional screening of the remaining abstracts to assess geographical distribution? Or that they conducted the initial screening of 99, and already found that the distribution was limited? Perhaps another arrow from the “abstracts screened” box directly to the “screening abstracts revealed…” box could clarify this in the figure?

Line 158: Table 2- Row 1 Column 3 -“Geographic location of the Study” - is the capitalization of “study” intentional? If so, disregard the comment. If not, de-capitalize.

Line 172: Table 2

What country did (29) get their funding from? At the moment the table states Asia.

(43) Could benefit from including China since not everyone would know what HSBC is. Assuming it is from China.

Figure 3: The Y-axis should be labeled.

Line 358: Table 4 Caption could use more detail for it to stand alone. e.g. “a summary of the …..of citizen science data tools identified in a scoping review of….”

Line 372: Can the authors indicate what “These” are? I suggest rephrasing to explicitly mention what they are referring to. E.g. “Proposals for data validation methods are evident…”

Ling 354: Section 3.7: This section could benefit from additional sub-headings such as Data collection + Data Validation and Verification. If three levels of headers are OK, please consider revising.

Line 381: The authors use et al on this line, can you consider using this naming in other parts of the manuscript where it can help with sentence flow? For example, Line 368 could flow better if it were Vattakaven et al.

Line 443: Thank you for acknowledging the non-English exclusion limitation. Another sentence that explains what this exclusion may have missed could be useful here.

Line 445: The authors state “that citizen science research in LMICs continues to be predominantly driven by institutions and funding agencies from the Global North…”, but the abstract only talks about the predominance of funding. Is there a reason for leaving out the institutions part there?

Line 512: Table 5 caption can be more detailed

Is this an exhaustive list of environmental changes? As I look at the figure with “Environmental changes and ecosystem impairment,” I am left to wonder if these were fully considered in the environmental changes list? Or if these indicators were just from the scoping review?

All of the categories could use the same level of definition that the authors provide for the “Direct health effects” category.

Grammar under the Human health outcomes list. eg should be e.g., - can this be applied throughout where it also applies?

For Demographics-Sex - could this exclude those who do not identify with a sex? Use of Examples may address this item (e.g., Male, Female, Other)

Discussion section considerations:

On the youth:

I note that several references involved youth and that the conclusion recommends " further research into how youth-driven citizen science initiatives can enhance and accelerate advancements in planetary health…” Given the conclusion's emphasis on youth, the findings and discussion could benefit from additional detail that justifies the recommendation.

On regional unrest:

The authors note that regional unrest (Line 423, 502) was an external factor that posed a barrier in a study. Given that several LMICs experience some form of instability, particularly political instability, is there any relevance to including literature on this observation in the discussion? Upon searching "citizen science" in "fragile" contexts with limited government” on Google Scholar, I came across these references:

Fan, F. ti, & Chen, S. L. (2019). Citizen, Science, and Citizen Science. East Asian Science, Technology and Society: An International Journal, 13(2), 181–193. https://doi.org/10.1215/18752160-7542643

Benyei, P., Skarlatidou, A., Argyriou, D., Hall, R., Theilade, I., Turreira-García, N., Latreche, D., Albert, A., Berger, D., Cartró-Sabaté, M., Chang, J., Chiaravalloti, R., Cortesi, A., Danielsen, F., Haklay, M. (Mordechai) ., Jacobi, E., Nigussie, A., Reyes-García, V., Rodrigues, E., Sauini, T., Shadrin, V., Siqueira, A., Supriadi, M., Tillah, M., Tofighi-Niaki, A., Vronski, N. and Woods, T. (2023) ‘Challenges, Strategies, and Impacts of Doing Citizen Science with Marginalised and Indigenous Communities: Reflections from Project Coordinators’, Citizen Science: Theory and Practice, 8(1), p. 21. Available at: https://doi.org/10.5334/cstp.514.

*I am not asking to cite these above, rather see if there is any literature that can support this observation.

6. PLOS authors have the option to publish the peer review history of their article (what does this mean? ). If published, this will include your full peer review and any attached files.

**Do you want your identity to be public for this peer review?** For information about this choice, including consent withdrawal, please see our Privacy Policy .

Reviewer #1: **Yes: ** Meghnath Dhimal

Reviewer #2: **Yes: ** Awah Kum Tchouaffi

Reviewer #3: No

Reviewer #4: **Yes: ** Gloria C. Blaise

---

## [Decision Letter · Decision Letter 1]

12 Mar 2025

PGPH-D-24-02515R1

Potential of citizen science to advance urban planetary health research in low and middle-income countries: A scoping review

Dear Dr. Tolu Oni,

Thank you for submitting your manuscript to PLOS Global Public Health. After careful consideration, we feel that it has merit but does not fully meet PLOS Global Public Health’s publication criteria as it currently stands. Therefore, we invite you to submit a revised version of the manuscript that addresses the points raised during the review process.

We look forward to receiving your revised manuscript.

Kind regards,

Muhammad Asaduzzaman, MD MPH MPhil

Academic Editor

Journal Requirements:

Additional Editor Comments (if provided):

Reviewers' comments:

Reviewer's Responses to Questions

**Comments to the Author**

1. If the authors have adequately addressed your comments raised in a previous round of review and you feel that this manuscript is now acceptable for publication, you may indicate that here to bypass the “Comments to the Author” section, enter your conflict of interest statement in the “Confidential to Editor” section, and submit your "Accept" recommendation.

Reviewer #5: (No Response)

Reviewer #6: (No Response)

2. Does this manuscript meet PLOS Global Public Health’s publication criteria ? Is the manuscript technically sound, and do the data support the conclusions? The manuscript must describe methodologically and ethically rigorous research with conclusions that are appropriately drawn based on the data presented.

Reviewer #5: Yes

Reviewer #6: Yes

3. Has the statistical analysis been performed appropriately and rigorously?

Reviewer #5: N/A

Reviewer #6: N/A

4. Have the authors made all data underlying the findings in their manuscript fully available (please refer to the Data Availability Statement at the start of the manuscript PDF file)?

Reviewer #5: (No Response)

Reviewer #6: Yes

5. Is the manuscript presented in an intelligible fashion and written in standard English?

Reviewer #5: Yes

Reviewer #6: Yes

6. Review Comments to the Author

Reviewer #5: Thank you for this excellent and very interesting paper which is a great contribution to the field. I particularly enjoyed the breakdown of funding sources and institutional dominance, the discussion of how involved people were and also the different degrees to which the studies tried to achieve change as part of the study - this felt novel and very interesting to situate the studies within their own context and power dynamics.

General comments

The paper frames itself through the lens of the ‘Anthropocene’. Anthropocene as a term has been criticised for laying the blame for ecological destruction with generic references to ‘human’ activities, rather than explicitly naming an extractive economy. In reality, the timeframe referred to as the Anthropocene coincides with accelerated industrial/colonial capitalism, not the presence of humans, I would suggest using a different term or caveating the term Anthropocene with a reference to extractivism/the extractive economy (defined in the Movement Generation Just Transition book, for example) rather than human activities throughout the introduction and the paper in general. Similarly, Figure 2 refers to an escalation of ‘human pressure’ - again this falls into the same trap of placing equal blame on ‘humans in general’.

Overall, the paper could benefit, in the introduction, analysis and conclusion, from increased discussion of the concept of accountability. Are the studies included seeking accountability? Are the issues being researched primarily imposed on the communities by global North corporations or other activities outside of the community? How is the burden of environmental harm disproportionate to resource use and contribution to the climate crisis? I think an increased consideration of accountability is extremely relevant given the focus on citizen science, where often communities are researching harm done to themselves and their environment for which they are not responsible, and given the references to planetary boundaries.

The analysis, recommendations and conclusion frequently refer to professional researchers, and the importance of them training and validating the research produced by communities. What is not discussed in this paper is a process of mutual learning - how can citizen science be a way for professional researchers to learn from community expertise? How can it be a methodology which expands the possibility of including Indigenous knowledge systems into scientific research? Perhaps this was not a factor in any of the studies, and they remained top-down, but this is worth discussing more in my opinion, perhaps in the conclusion and recommendations if it was not part of the studies you found.

Specific comments

24: ‘These settings are under going rapid urbanisation that would further threaten planetary boundaries’ feels out of sync with the proportion of responsibility for expanding planetary boundaries, I would suggest reframing this sentence regarding the specific health risks of rapid urbanisation and the ‘exposures mentioned’ e.g. pollution, overcrowding, overheating, etc, rather than centring planetary boundaries.

49: Exceeding planetary boundaries is not the only threat to human health. It is the process of exceeding those boundaries - i.e. the creation of pollution, deforestation and other ecologically destructive activities that harm human health. The introduction would benefit from mentioning the health relationships of the different planetary boundary indicators and the processes by which they are overshot, and clarifying the process of exceeding planetary boundaries also destroys health. Similarly, figure 2 focusses on the consequences, not the process of environmental degradation harming human health e.g. through oil spills and gas flaring which then leads to Co2 rise and the health harms of climate change. This is particularly relevant for citizen science which often measures the process e.g. of pollution, not only the outcome, of planetary change. A 'cradle to crave' analysis of extractivism may be helpful here - there is a Global Climate and Health Alliance report on this for the oil and gas industry which could be a helpful reference.

75: This paragraph cites the popularity of citizen science primarily as a result of its ability to collect large datasets and produce high quality research. I think it would be interesting to reference the history of citizen science as a necessity and a site of resistance and agency - where communities and activists have build their own research practices because public health has invalidated their experiences, especially Indigenous communities and communities affected by environmental racism.

90: Similar comment re rapid urbanisation being implied as the main driver of environmental degradation in LMICS. Whilst environmental degradation is hugely and disproportionately accelerated in the global South, this paper would benefit from widening the scope of the the causes it sites for that reason, in particular referencing colonialism, neocolonialism, extractive industries, the electric/precious metals scramble, oil and gas expansion, and unequal climate impacts, for example.

139: I would like to know more about the specific search terms / search strings / key words used in the study search, and more information as to what is classified as a ‘planetary health relevant study’ or ‘citizen science projects that focus on promoting planetary health’ an ‘empirical study on citizen science relevant to planetary health in urban settings of LMICs’ etc. Given that planetary health is a relatively new and field-specific term, there is a risk that this paper has excluded large areas of citizen science depending on which terms were used. In particular, I am thinking of the long history of the use of citizen science in the environmental justice movement. The paper would benefit from clarifying if environmental justice and associated search terms (such as environmental racism, Indigenous science, pollution) were incorporated with the keyword of citizen science, and reflection on if that affected the number of results.

483: Is it worth noting that none of the studies (as far as I can tell) measured health outcomes alongside environmental indicators such as water quality, biodiversity and air quality. The studies which measure human health outcomes do it in isolation, as do the studies which measure environmental indicators. I think it could be worth discussing and including in the conclusion - what would it look like to do citizen science that combined both exposure e.g. flooding, with peoples self reported health symptoms? How could tools like this be validated? What could this mean for seeking change?

490: This discussion of political instability/political intervention would also benefit from a discussion re accountability - how does political instability intersect with the process of pollution and environmental extraction, therefore making the citizen science that could show that harm harder? Who benefits from the political instability making citizen science harder? Did any of the papers incorporate safety measures or other protection for citizen scientists? What challenge does citizen science pose to powerful interests and what are the implications for how to expand it?

552: I would expand the environmental changes of planetary significance to be more specific of environmentally destructive activities such as deforestation, soil erosion, mining etc, rather than generic changes. I would add noise pollution to direct health effects and expand injury to include direct violence. You could potentially also add carcinogen exposure or cancer to direct health effects. I would expand aesthetics and culture to be more specific or give some examples, such as loss of traditional territory or access to sacred sites. I would add poverty and loss of livelihood. I would also include a reporting indicator on purpose of the study / each party’s motivation in the study e.g. policy change. I would also include a reporting indicator on what safety, wellbeing check ups and support are available to citizen scientists.

566: Majority World is a great term but only appears at the end of the paper - maybe could include at the beginning when you first introduce the term LMIC and suggest it as an alternative and then use consistently throughout the paper?

600: I think the conclusion could be stronger regarding the potential of citizen science to fundamentally try to halt, change and repair environmental damage and planetary harm, not only to observe it. How can citizen science be a tool for empowering, supporting and engaging communities to in the challenges they face? How can it be a way to relate to a rapidly changing ecological system, and through that relationship repair the land and our own health?

Reviewer #6: I appreciate the significant time and effort it took to gather this information. This manuscript has already gone through a round of reviews and edits/responses, which I will defer to those reviewers if they are satisfied with the authors' responses. I will make some other general comments:

I do find this paper quite useful as a global health practitioner and environmental scientist. I particularly appreciate tables 5 and 6 with useful recommendations for standardization. I also appreciate the authors' defense of their methodologies. I realize that some of my comments may be outside of what can now be gleaned from the data, but I will still mention some areas I would be interested in more clarification if the authors/editors choose to explore them:

1) what specific search terms were used? JUST citizen science for example, or were there synonyms for that term used. Same for "planetary health" or boundaries, etc. I know it says synonyms were used, but I am unclear on this and I would be curious to know those terms. If not, there may be studies absent from the review that termed their methods something different but with the same spirit of community involvement in data collection. I would be curious also to be able to conduct the search as a reader.

2) Did the included studies all conduct analysis on efficacy or were some only descriptions of project methodologies and how to engage the community in this work?

3) I would be very curious to know more about the contexts for these studies.. WHY did they choose the citizen science methodology over other designs? Was it for resource reasons? For the sake of community engagement itself? For promotion of awareness in a community about an environmental issue? This may be too late to address in this manuscript, but as a global health/environmental health scientist working in LMICs, this would be a very useful addition for me.

4) What were the criteria for the studies being planetary health related? I see they were agreed upon by reviewers, but what were they. I see that some studies appear to only address a health topic (example: cvd risk), while others strictly environmental, while some had more obvious crossover. Were all studies related to health OR environmental topics included as long as they used citizen science methodologies - or were there very specific things that made them fall into the "Planetary Health" category?

5) I do agree with some of the previous reviewers that the discussion could be more in depth, but I realize the paper is already quite long and the authors are clear that their intent was more in aggregation of the information for future in-depth analysis. I would however love to see some discussion on the efficacy of this methodology. How much better or how much more data, or whatever the "more" positive outcome may be for the study by using citizen science methodology? What are the authors' impressions from the over 30 studies they analyzed? Is this a worthwhile investment of resources??

7. PLOS authors have the option to publish the peer review history of their article (what does this mean? ). If published, this will include your full peer review and any attached files.

**Do you want your identity to be public for this peer review?** For information about this choice, including consent withdrawal, please see our Privacy Policy .

Reviewer #5: **Yes: ** Dr Rhiannon Mihranian Osborne

Reviewer #6: No

---

## [Editor Report · Decision Letter 2]

24 Mar 2025

Potential of citizen science to advance urban planetary health research in low and middle-income countries: A scoping review

PGPH-D-24-02515R2

Dear Tolu Oni,

We are pleased to inform you that your manuscript 'Potential of citizen science to advance urban planetary health research in low and middle-income countries: A scoping review' has been provisionally accepted for publication in PLOS Global Public Health.

Best regards,

Muhammad Asaduzzaman, MD MPH MPhil

Academic Editor